# LoopGaussian: Creating 3D Cinemagraph with Multi-view Images via Eulerian Motion Field

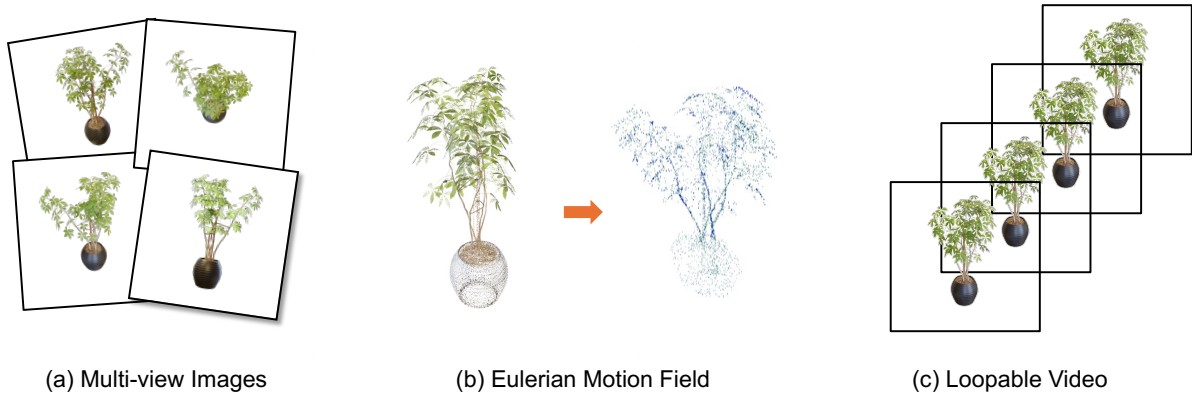

(a) Multi-view Images  (b) Eulerian Motion Field  (c) Loopable Video

**Figure 1:** We propose **LoopGaussian**, a novel method designed to convert multi-view images of a stationary scene (a) into authentic 3D cinemagraph by an Eulerian motion field (b). The 3D cinemagraph can be rendered from a novel viewpoint to obtain a natural seamless loopable video (c).

## ABSTRACT

Cinemagraph is a unique form of visual media that combines elements of still photography and subtle motion to create a captivating experience. However, the majority of videos generated by recent works lack depth information and are confined to the constraints of 2D image space. In this paper, inspired by significant progress in the field of novel view synthesis (NVS) achieved by 3D Gaussian Splatting (3D-GS), we propose **LoopGaussian** to elevate cinemagraph from 2D image space to 3D space using 3D Gaussian modeling. To achieve this, we first employ the 3D-GS method to reconstruct 3D Gaussian point clouds from multi-view images of static scenes, incorporating shape regularization terms to prevent blurring or artifacts caused by object deformation. We then adopt an autoencoder tailored for 3D Gaussian to project it into feature space. To maintain the local continuity of the scene, we devise SuperGaussian for clustering based on the acquired features. By calculating the similarity between clusters and employing a two-stage estimation method, we derive an Eulerian motion field to describe velocities across the entire scene. The 3D Gaussian points then move within the estimated Eulerian motion field. Through bidirectional animation techniques, we ultimately generate a 3D Cinemagraph that exhibits natural and seamlessly loopable dynamics. Experiment

results validate the effectiveness of our approach, demonstrating high-quality and visually appealing scene generation.

## CCS CONCEPTS

• **Computing methodologies** → **Computer graphics**; *Image-based rendering*; *Image and video acquisition*;

## KEYWORDS

Cinemagraph, Dynamic scene generation, 3D scene reconstruction

**ACM Reference Format:**
Anonymous Authors. 2024. LoopGaussian: Creating 3D Cinemagraph with Multi-view Images via Eulerian Motion Field. In *Proceedings of Make sure to enter the correct conference title from your rights confirmation emai (Conference acronym 'XX)*. ACM, New York, NY, USA, 10 pages. https://doi.org/XXXXXXX.XXXXXXX

## 1 INTRODUCTION

*Cinemagraphs* are static images in which a minor and repeated movement appears, forming a seamlessly looping video clip [10]. It offers a unique blend of static imagery and subtle motion by combining elements of both photography and videography, captivating audiences with its mesmerizing allure. In light of recent advancements in concepts such as augmented reality, mixed reality, and metaverse, there has been a growing demand for creating natural and realistic cinemagraphs [2, 19]. However, generating dynamic 3D scenes within cinemagraphs typically entails extensive manual labor from professional artists, leading to considerable costs.

Existing works [12, 14, 30] mainly focus on automatically creating cinemagraphs from static images in 2D image space. Mahapatra and Kulkarni [30] propose to interactively animate fluid elements

in still images based on flow directions and regions of interest provided by users. However, these manipulations are all done in 2D image space where the view direction is inevitably fixed, which lacks a sense of visual fidelity.

A few methods [19, 29] that explore the generation of 3D cinemagraphs have recently emerged. For example, Li et al. Li et al. [19] take the first step to create realistic animations of scenes with compelling parallax effects by jointly learning image animation and novel view synthesis. Ma et al. Ma et al. [29] introduce a pipeline to create cinemagraphs from asynchronous multi-view videos, thereby facilitating the exploration of diverse viewpoints. Nevertheless, we argue that these approaches cannot be considered as *authentic* 3D cinemagraphs, as they merely employ representations of multi-layer two-dimensional images with depth information such as MPI [55] or LDI [44], failing to reconstruct the underlying three-dimensional geometric structure of scenes. Consequently, these methods struggle to produce the effect of camera movement or need to restrict the camera movement within a confined viewing angle and range. In addition, the lack of geometric information may lead to artifacts or produce geometric inconsistency.

In consideration of these constraints in existing methodologies, we present *LoopGaussian*, an innovative framework for curating *authentic* 3D cinemagraphs from multi-view images of static scenes. LoopGaussian is grounded in the reconstruction of the 3D structure of the scene from multi-view images, taking advantage of the state-of-the-art 3D Gaussian Splatting [17]. Consequently, we can efficiently perform operations that are previously unattainable in 2D space, such as novel view synthesis and dynamic scene rendering. Moreover, our method can effectively leverage the inherent self-similarity within the scene representations, eliminating the necessity for pre-training on extensive datasets.

We streamline our framework into three steps, as illustrated in Fig. 1: (a). Firstly, we train a 3D Gaussian scene using multi-view static images, followed by an autoencoder that maps 3D Gaussian points into an appropriate feature space. (b) Next, we devise a clustering method on these 3D Gaussian points and exploit self-similarity among clusters to construct a velocity field. The velocity field is further refined with a multi-layer perceptron to obtain an Eulerian motion field. (c) Finally, we generate a seamlessly loopable video based on the derived Eulerian motion field with bidirectional animation techniques. The generated video clip is more plausible and visually captivating owing to its generation process in the three-dimensional space. In our experiments, we consider simulating motions of soft and non-rigid objects, such as tree branches, flags, and hanging clothes. Our experimental results demonstrate the effectiveness of our proposed method. In summary, our main contributions are as follows:

- We propose a novel framework capable of generating authentic 3D Cinemagraphs from multi-view images of static scenes, which achieves seamless loopable dynamics of the scene and can be rendered from a novel viewpoint.
- We innovatively describe the dynamics of the scene in terms of Eulerian motion fields in 3D space. Leveraging the scene's self-similarity, we employ a two-stage optimization strategy to estimate the Eulerian motion field.

- Our framework is heuristic, obviating the necessity for pre-training on large datasets, and it offers flexibility by enabling users to control the magnitude of the scene dynamics.

## 2 RELATED WORK

### 2.1 Cinemagraph

A *cinemagraph* [10, 43] is a combination of a static image and a dynamic video, where most of the scene is still while a fraction of it changes in a continuous loop. This concept has gained popularity in diverse domains, ranging from artistic expression and digital storytelling to advertising and brand marketing. While advanced digital tools have empowered artists and photographers to craft photo-realistic cinemagraphs, the manual creation process is still labor-intensive and time-consuming.

There exists a rich body of works [1, 21, 22, 34, 45, 52] that explore an automatic creation of cinemagraphs. Earlier methods commonly take a video as input to generate a seamlessly looping video clip. For instance, by identifying segments with cyclic motion properties [21, 45, 52]. Agarwala et al. [1] create panoramic video textures from the output of a single panning video camera. Oh et al. [34] introduce an end-to-end approach that extracts high-level semantics from the input video to facilitate cinemagraph generation. In addition, there are several works [6, 7, 23] that aim at creating a cinemagraph from a single image. Chuang et al. [7] propose a semi-automated method that lets users manually segment the scene into multiple layers. Subsequently, stochastic motion textures are automatically synthesized for each layer, which are then integrated to create the final video. Lin et al. [23] demonstrate the capability of generating waterfall animations from static waterfall images. Choi et al. [6] automatically generate cinemagraphs from a still landscape image using a pre-trained StyleGAN [16]. Text2Cinemagraph [31] presents a fully automatic method that synthesizes cinemagraphs from text descriptions. However, these works are all constricted to 2D image space, which may fail to deliver an immersive experience for audiences.

Apart from 2D cinemagraph generation, Li et al. [19] pioneered a framework for generating 3D cinemagraphs from a single still image. This approach utilizes a dense depth map to separate the scene into several layers and expand 2D motion into 3D scene flow. In contrast, we directly handle 3D points derived from the scene in 3D space without endeavor of predicting any depth maps, yet still achieving visually appealing looping effect. Moreover, our method is capable of exploring representation similarity in 3D space to construct cinemagraphs, thus alleviating the need for pre-training on large-scale datasets.

### 2.2 Neural Scene Representation

Neural scene representation aims at modeling the scene via neural networks, in which way the entire rendering pipeline is differentiable, and the whole scene is learnable. Neural Radiance Field (NeRF) [32] is one of the most popular methods for implicit neural scene representation. It models the view-dependent color and opacity at each spatial position through Multi-Layer Perceptrons (MLPs), and enables novel view synthesis through volume rendering. The

success of NeRF has inspired a significant body of subsequent research. Some studies are devoted to improving the training and rendering speed [13, 27, 33, 41, 53], while others aim at elevating the rendering quality [3, 18, 42, 47, 48]. Additionally, several efforts are made towards adapting NeRF for dynamic scenes [4, 11, 37, 39].

3D Gaussian Splatting (3D-GS) [17] is an emerging neural scene representation approach that explicitly models 3D information of a scene. It employs a collection of semi-transparent anisotropic Gaussian ellipsoids to represent the input scene, and designs a differentiable rasterization rendering pipeline to enable real-time high-fidelity rendering. While 3D-GS is initially designed for static scenes, some subsequent works have extended it to handle dynamic scenes. One line of work is centered around the idea of predicting the temporal evolution of individual 3D Gaussians using neural networks [8, 15, 24, 49, 51], while another entails representing a dynamic scene within a 4D space where each moment is represented as a slice of this space [9, 20, 28]. Although 3D-GS achieves notable success in modeling dynamic scenes, existing approaches are generally based on the Lagrangian perspective. In this paper, we make an early attempt to describe the progressive dynamics of a scene from an Eulerian perspective.

## 3 PRELIMINARIES

### 3.1 3D Gaussian Splatting

3D Gaussian Splatting (3D-GS) [17] has recently emerged as an explicit scene representation and rendering approach, which allows high-quality and real-time rendering for scenes captured with images from *multiple* viewing directions.

3D-GS explicitly represents the scene as a set of 3D Gaussian points. Each 3D Gaussian point $G_i$ is characterized with attributes $\{p_i, q_i, s_i, \sigma_i, c_i, SH_i\}$, where $p_i \in \mathbb{R}^3$ is the space position, $q_i \in \mathbb{R}^4$ is the quaternion representing rotation, $s_i \in \mathbb{R}^3$ is the scaling on each axis, $\sigma_i \in \mathbb{R}$ is the opacity, $c_i \in \mathbb{R}^3$ is the diffuse color, and $SH_i$ is the spherical harmonic function to express anisotropic colors. The dimensions of $SH$ are depended on the order used. The shape of each 3D Gaussian point is controlled by a covariance matrix $\Sigma = RSS^T R^T$, where $R$ is the rotation matrix transformed from quaternion $q$, and $S$ is the scaling matrix transformed from $s$. Hence, each 3D Gaussian point can be expressed as:

$$G(x) = e^{-\frac{1}{2}(x-p)^T \Sigma^{-1}(x-p)}. \qquad (1)$$

During the rendering procedure, each 3D Gaussian is projected onto an image plane in camera space to shape a 2D Gaussian. To determine the color of each pixel, the Gaussians that are contained in one pixel are sorted by depth, and the pixel color $\hat{c}$ is estimated according to $\alpha$-blending:

$$\hat{c} = \sum_{i \in G_{\text{pixel}}} \hat{c}_i \hat{\alpha}_i \prod_{j=1}^{i-1} (1 - \hat{\alpha}_j), \qquad (2)$$

where $G_{\text{pixel}}$ is a set of Gaussians that are contained in this specific pixel, $\hat{c}_i$ is the learned color of the Gaussian, and $\hat{\alpha}_i$ is the learned opacity multiplied with the Gaussian.

### 3.2 Eulerian Motion Field

Existing approaches that capture Gaussian point cloud dynamics are generally based on the Lagrangian perspective. The Lagrangian methods track individual particles over time, focusing on their trajectories as they move through space, which is fairly intuitive. In contrast, the Eulerian perspective focuses on specific locations in space, observing how particles move through these locations over time. To approximate such an Eulerian motion field, Holynski et al. [14] propose a static motion field in 2D image space, where the value of each pixel defines its immediate velocity that remains constant over time. We follow this method and adapt it to describe the deformation of soft non-rigid objects (such as branches, flags, ropes, etc.) in 3D space. Formally, the motion of a particle from one frame to the next through Euler integration is described as follows:

$$X(t + 1) = X(t) + \vec{E}(X(t)), \qquad (3)$$

where $\vec{E}$ is the static Eulerian motion field and $X(t)$ is the position of a particle at time $t$.

## 4 METHOD

### 4.1 Overview

Given a set of multi-view images of a static scene, our goal is to create a seamlessly loopable and natural-looking 3D cinemagraph. The overview of our method is illustrated in Fig. 2. We start by creating a 3D Gaussian point cloud using 3D-GS [17] with an additional eccentricity regularization (Sec. 4.2). As not all objects in the scene are suitable for deformation, we segment the parts that are likely to exhibit loopable motion with SAGA [5]. The dynamic 3D Gaussians are then projected into a feature space via an autoencoder and clustered according to both position and feature information through our designed SuperGaussian approach (Sec. 4.3). Next, we derive a global feature for each cluster and calculate the similarity among clusters based on these global features. The similarity information is used to estimate a velocity field, which is further refined with an MLP as the final Eulerian motion field (Sec. 4.4). Finally, we achieve a loopable video based on our estimated Eulerian motion field using bidirectional animation technology [14, 19, 30] (Sec. 4.5). We will elaborate on the details in the following subsections.

### 4.2 Artifact-free Scene Representation

***3D Gaussians Generation.*** We obtain the flexible representation of the initial static scene, which is composed of 3D Gaussian points, using 3D-GS [17] as described in Sec. 3.1. We choose 3D-GS because a 3D Gaussian point is essentially an ellipsoid, the shape of which can efficiently and accurately characterize intricate geometries when the scene is static. However, the dynamics of a scene's deformation can inevitably modify the positional relationships among these Gaussian points, and the excessively sharp ellipsoids may introduce glitch artifacts, thereby compromising the visual coherence of the scene. To address this, inspired by prior works [26, 50], we improve the representation robustness and visual fidelity of the scene by introducing a constraint on shape during training. Specifically, we incorporate a regularization term targeting at *ellipsoidal*

**Figure 2: Overview of our framework.** Given multi-view images of a static scene, we initially create a 3D Gaussian point cloud using 3D-GS with an eccentricity regularization term. Next, we identify the point cloud region that the user wishes to deform using a 2D Mask. The 3D Gaussians are then projected into the feature space via an autoencoder and undergo clustering using SuperGaussian. Subsequently, we derive a sparse velocity field based on self-similarity, interpolate to acquire a dense velocity field and refine the final Eulerian motion field through an MLP. Finally, we can generate a seamlessly loopable video by leveraging bidirectional animation techniques in 3D space and incorporating specified camera parameters.

*eccentricity* when training the 3D Gaussian scene:

$$\mathcal{L}_{\text{shape}} = \frac{1}{|\mathbf{G}|} \sum_{G_i \in \mathbf{G}} 1 - \frac{\min^2(s_i)}{\max^2(s_i)}, \qquad (4)$$

where $s_i$ is the scaling on each axis and $\mathbf{G} \coloneqq \{G_i\}$ is the set of 3D Gaussian points of the scene as described in Eq. (1).

**3D Gaussians Optimization.** During the training process, the 3D Gaussian point cloud is rendered to a novel view image through the rasterization pipeline explained in Sec. 3.1. The error is then calculated as the difference between the rendered image and its corresponding ground-truth image. Following [17], we adopt the absolute error $\mathcal{L}_1$ and the structural similarity index $\mathcal{L}_{\text{D-SSIM}}$ as the difference measure. Mathematically, the total loss for 3D Gaussians optimization is defined as:

$$\mathcal{L}_{\text{3D-GS}} = \eta \left( (1 - \beta) \, \mathcal{L}_1 + \beta \mathcal{L}_{\text{D-SSIM}} \right) + (1 - \eta) \, \mathcal{L}_{\text{shape}}, \qquad (5)$$

where $\mathcal{L}_{\text{shape}}$ is our introduced regularization loss on ellipsoidal eccentricity, $\beta$ is a weighting factor that balances $\mathcal{L}_1$ and $\mathcal{L}_{\text{D-SSIM}}$, and $\eta$ is another weighting factor that balances the error loss and the regularization loss.

Compared with the latest works on cinemagraph generation, the benefits of representing the scene with 3D Gaussians and optimizing it using Eq. (5) are three-fold. First, we can take as input multiple images from different viewing directions instead of just one single image, which is advantageous for reconstructing the intricate 3D geometries of the observed scene. Second, we present an auxiliary regularization term on the shape of 3D Gaussians to mitigate the artifact issue, which will be empirically validated in Sec. 5.4. Third,

we can efficiently construct the subsequent Eulerian motion field by exploiting the distance relationships among 3D Gaussians.

**3D Gaussians Separation.** After learning the reconstruction of the 3D Gaussian point cloud, we manually annotate the objects that are anticipated to have motion effects within images in the training set. Next, we utilize SAGA [5], an interactive segmentation approach for 3D Gaussians, to create a mask for the point cloud. This mask is introduced to segregate the 3D Gaussian point cloud into static and dynamic components. Concretely, let $I$ represent the multi-view images of the training set and $A$ the manual annotations on these images, the binary mask $\mathbf{M}$ is obtained by:

$$\mathbf{M} \coloneqq \{m_i\} = \text{SAGA}(\mathbf{G}, I, A) \qquad (6)$$

For each Gaussian point $G_i$, if its corresponding mask value $m_i$ is 1, it will participate in the subsequent construction of Eulerian motion field; otherwise, it remains stationary. This separation process offers a flexible way to concentrate exclusively on modeling the dynamic 3D Gaussians. For brevity, we reuse the notation $\mathbf{G}$ to denote *dynamic* 3D Gaussians in the next subsections.

## 4.3 SuperGaussian for 3D Gaussians Clustering

The point cloud of 3D Gaussians offers an unstructured representation of a scene; however, the scene it depicts usually exhibits a *structured* geometry. An illustrative example is observing a flag waving in the wind: if a corner of the flag moves in a specific direction, the entire flag is likely to exhibit similar motion patterns. This local coherence of motion originates from the physical interconnection within the flag. Generally, if one point of an object moves,

the surrounding points may also show a similar trend of movement due to the local consistency of geometry.

In consideration of the analysis above, we borrow the concept from *supervoxel* to preserve the local consistency of geometry. The basic idea of supervoxels involves clustering 3D points according to spatial proximity and feature similarity. Inspired by previous works on supervoxel segmentation [25, 36], we introduce a clustering method for 3D Gaussians, termed *SuperGaussian*. Concretely, let $\mathbf{G} \coloneqq \{G_i | i = 1, \cdots, N\}$ be the set of 3D Gaussians in a point cloud and $\mathbf{C} \coloneqq \{C_k | k = 1, \cdots, K\}$ the set of clusters, where $N$ is the total number of points and $K$ is total number of clusters. We first partition the scene into voxels based on a voxel resolution $R$, and then randomly select a *seed* Gaussian point within each of the non-empty voxels (i.e., voxels that contain at least one Gaussian point). Let $\text{SG}(\cdot)$ represent the SuperGaussian model that assigns a clustering label to a Gaussian point, and $\text{SG}^*$ the optimized Super-Gaussian. Clustering is achieved through the optimization of the following objective function:

$$\text{SG}^* = \underset{\text{SG}}{\arg\min} \sum_{k=1}^{K} \sum_{\text{SG}(G_i)=k} D(G_i, G_{k'}), \qquad (7)$$

where $k'$ is the corresponding index of the seed Gaussian point for cluster $k$. The metric function $D(\cdot, \cdot)$ is defined as:

$$D(G_i, G_j) = 1 - \frac{|f_i \cdot f_j|}{\|f_i\| \cdot \|f_j\|} + \mu \frac{\|p_i - p_j\|}{R}, \qquad (8)$$

where $f_i$ and $f_j$ are the features of the Gaussian points $G_i$ and $G_j$ from an autoencoder [10, 40], $p_i$ and $p_j$ are the positions of the Gaussian points, $\mu$ is a weighting factor that balances the importance of features and positions, and $R$ stands for the resolution of supervoxels. With the seed points selected as centers, we gradually search outwards, applying the metric function $D(\cdot, \cdot)$ to identify the appropriate cluster for each point. This iterative process continues until all 3D Gaussian points are successfully assigned to a cluster.

SuperGaussian can be seen as a variant of k-means, but enjoys the following advantages. On one hand, the seeding procedure is implemented by selecting seed Gaussian points in each non-empty supervoxels, so that the seeds are almost uniformly distributed across the scene. On the other hand, we incorporate the attributes of the learned 3D Gaussians into the distance measure, which ensures that the clustering algorithm will converge in just a few iterations. We experimentally find that only one iteration is enough to achieve a satisfactory clustering result, which makes SuperGaussian even more efficient.

## 4.4 Progressive Eulerian Motion Field Estimation

Motion field estimation in natural scenes is a challenging task due to the scarcity of comprehensive datasets, particularly for point clouds. A significant limitation arises from the difficulty in predicting subsequent scene flow based solely on static point cloud data. In response to this challenge, we propose a hypothesis: Similar objects generally have similar movement trends. The rationale behind this hypothesis stems from the observation that objects within natural scenes often exhibit collective behavior or tend to interact with one another in predictable ways. For instance, when a breeze blows through a forest, leaves on nearby trees tend to move in unison, following a similar direction. Building upon this hypothesis, we devise an efficient estimation method for the Eulerian motion field according to the similarity relationships among clusters.

***Sparse Velocity Field Estimation.*** We derive an initial velocity field by moving each cluster to its nearest neighbor. To achieve this, we first need to identify a global feature for each cluster, on which a similarity metric can be performed. Considering the disorder of Gaussian points in each cluster, we adopt maximum pooling, a permutation-invariant and symmetric function, to extract the global feature $f_{C_i}$ for each cluster $C_i$. Next, we obtain a similarity matrix $\mathbf{S} \coloneqq \{s_{ij}\}$ by computing the cosine similarity between two global features:

$$s_{ij} = \frac{f_i \cdot f_j}{\|f_i\| \cdot \|f_j\|} \quad \forall i \in [1, K], j \in [1, K]. \qquad (9)$$

The most similar cluster $C_{\tilde{i}}$ for each cluster $C_i$ is then identified with the highest cosine similarity:

$$j^* = \underset{j \neq i}{\arg\max}(s_{ij}) \quad \forall i \in [1, K]. \qquad (10)$$

To create the velocity field, we define a *center* point with position $\bar{p}_i$ for each cluster, which is calculated by taking the average of positions of all Gaussian points within the cluster. The velocity field $\mathbf{v}_{\text{sparse}} \coloneqq \{v_{s_i}\}_{i=1}^{K}$ is then achieved by moving one cluster to another with the highest similarity:

$$v_{s_i} = \bar{p}_{j^*} - \bar{p}_i \quad \forall i \in [1, K]. \qquad (11)$$

Note that the initial velocity field is fairly *sparse* due to the limited number of clusters. This sparse velocity field is prone to overfitting as they lack the granularity necessary for smooth representation across the entire spatial domain. This limitation may become more prominent when the motion field extends into unfamiliar areas, which can greatly increase the risk of divergence and discontinuity. Next, we will introduce our solution to this issue.

***Dense Velocity Field Estimation.*** We opt for Kriging interpolation [35] to estimate a *dense* velocity field from the sparse one calculated in Eq. (11). Kriging is a geostatistical interpolation method that is widely used for estimating the value of a variable at an unmeasured location based on the values of neighboring positions with known values.

Concretely, we compute the velocity at each position of a 3D Gaussian point via Kriging interpolation, which results in a dense velocity field $\mathbf{v}_{\text{dense}} \coloneqq \{v_{d_i}\}_{i=1}^{N}$. The dense velocity $v_{d_i}$ at position $p_i$ is a weighted sum of the observed sparse velocities $\mathbf{v}_{\text{sparse}}$, where the weights are derived from a variogram. In practice, we employ a standard spherical model to solve the variogram. The Kriging interpolation procedure excels in preserving the smoothness of the velocity field, and we will empirically demonstrate the effectiveness of it in Sec. 5.4.

***Eulerian Motion Field Estimation.*** As the Eulerian motion field captures velocities at all spatial positions, it is crucial to ensure smoothness to prevent scene tearing during deformation.

To this end, we adopt an MLP to estimate the Eulerian motion field $\vec{E}_G$ for enhanced smoothness. The inputs to the MLP are the spatial positions $\{\bar{p}_i\}_{i=1}^{K}$ and $\{p_i\}_{i=1}^{N}$, and it predicts the velocities

at the corresponding positions. The training process of MLP is fully supervised by the sparse and dense velocities $\mathbf{v}_{\text{sparse}}$ and $\mathbf{v}_{\text{dense}}$. During inference, we can estimate the velocity $v = \vec{E}_G(p) := \text{MLP}(p)$ at any given position $p$ in 3D space via the Eulerian motion filed.

## 4.5 Loopable Dynamics Generation

Given the estimated Eulerian motion field $\vec{E}_G$, we can efficiently ascertain the motion of each 3D Gaussian point. However, the convergence of each point's movement within the Eulerian field may not be guaranteed over time. This is because an Eulerian field is not necessarily an irrotational field, which may lead to unexpected scene tearing. To address this challenge, inspired by the bidirectional animation technology [14, 19, 30] in image space, we make an effort to escalate it from 2D space to 3D space, ensuring that each point stays within a reasonable range of motion and will ultimately revert to its initial position.

For each 3D Gaussian point $G_i$, we introduce a vector $\psi \in \mathbb{R}^3$ to regulate the magnitude of motion on each axis. The vector is simultaneously controlled by the lengths of sides of the scene's axis-aligned bounding box $h \in \mathbb{R}^3$, the number of video frames $T$, and a hyperparameter $\omega$ that controls the amplitude of motion, i.e., $\psi = \frac{\omega}{T} \cdot e^{-h}$. The position $p_i(t)$ of the point $G_i$ at time $t$ is calculated through Euler integration from 0 to $t$:

$$p_i(t) = p_i(0) + \sum_{\tau=0}^{t-1} \psi \odot \vec{E}_G\left(p_i\left(\tau\right)\right), \tag{12}$$

$$\text{where} \quad p_i(\tau) = p_i\left(\tau-1\right) + \psi \odot \vec{E}_G\left(p_i\left(\tau-1\right)\right).$$

Here $\odot$ signifies the Hadamard product. To achieve forward and backward animation in 3D space, we calculate the spatial positions $p_i(t)$ and $p_i(t-T)$ of each 3D Gaussian point $G_i$ at frames $t$ and $t - T$ using Eq. (12). Then we derive its final position $\hat{p}_i(t)$ at time $t$ through linear interpolation:

$$\hat{p}_i(t) = \alpha p_i(t) + (1 - \alpha)p_i(t - T), \tag{13}$$

where $\alpha = (1 - \frac{t}{T})$. In this way, we obtain a temporal sequence of positions for the 3D Gaussian point cloud. Given the camera parameters, including its position and viewing angle, we employ the rendering pipeline outlined in Sec. 3.1 to render each frame, which ultimately forms a seamlessly looping video.

## 5 EXPERIMENTS

### 5.1 Datasets

We utilize a combined synthetic dataset to comprehensively evaluate our proposed method. Part of the dataset is from the static NeRF synthetic dataset [32]. Due to the scarcity of dynamic datasets depicting natural scenes, we also produce a brand new dataset that showcase dynamic nature scenes using Unity. This dataset follows the structure of the NeRF synthetic dataset and is accompanied by a corresponding synchronized ground truth video, which has a resolution of $900 \times 900$ pixels and consists of 48 frames. Noted that the video is non-loopable.

**Table 1: Quantitative comparison of average optical flow maps.**

| Methods | PSNR↑ | SSIM↑ | LPIPS↓ |
|---------|-------|-------|--------|
| Li [19] | 22.959 | 0.915 | 0.233 |
| Ours | **24.868** | **0.928** | **0.208** |

**Table 2: Quantitative comparison of generated videos.**

| Methods | FVD↓ |
|---------|------|
| Li [19] | 1174.948 |
| Ours | **933.824** |

### 5.2 Implementation Details

Our experiments are completed on a single NVIDIA GeForce RTX 4090 with the PyTorch framework [38]. We empirically set $\beta = 0.2$ and $\eta = 0.9$ in Eq. (5), and the total number of training iterations is $50,000$. The autoencoder that projects 3D Gaussian points into feature space is designed based on PointNet [40], where the architecture of the decoder is symmetric to that of the encoder. During the clustering of 3D Gaussian points, we adjust the scene resolution $R$ according to $\lambda \max(h)$ to balance granularity and detail preservation, where $\lambda = 0.04$ and $\mu = 0.5$ in Eq. (8). The MLP utilized to depict the Eulerian motion field has two hidden layers of size 128 and 64 respectively, with positional encoding applied to the input. For the amplitude control of motion, we set $\omega = 1.2$. Regarding final video rendering, we set the video duration $T$ to 48 frames, with a resolution specified as $900 \times 900$.

### 5.3 Results

To the best of our knowledge, we present the *first* work that creates cinemagraphs in the authentic 3D space. For fair comparison with state-of-the-art work on synthesizing cinemagraphs in 2D image space, here we render our cinemagraphs in 2D image space as well. However, we remark that our proposed method is more advanced, in the sense that our method is performed in 3D space and is capable of rendering from any viewpoint, which is impossible for any of previous related works.

***Quantitative Evaluation***. We perform a quantitative evaluation on the self-produced synthetic dataset mentioned in Sec. 5.1. To rigorously evaluate the effectiveness of our approach, we conduct a quantitative comparison between optical flow maps generated by our method and that of a reference video. Optical flow maps serve as a fundamental tool in video analysis, offering insights into the dynamic changes occurring both temporally and spatially within a video sequence. We commence the evaluation process by analyzing the optical flow starting from the second frame onwards. The optical flow between each frame and its preceding frame is computed and averaged to obtain a comprehensive optical flow map. We assess the quality of results using three popular image quality assessment metrics, i.e., PSNR, SSIM, and LPIPS [54]. The quantitative comparison results of the average optical flow maps are presented in Table 1. It is evident from the table that our method's

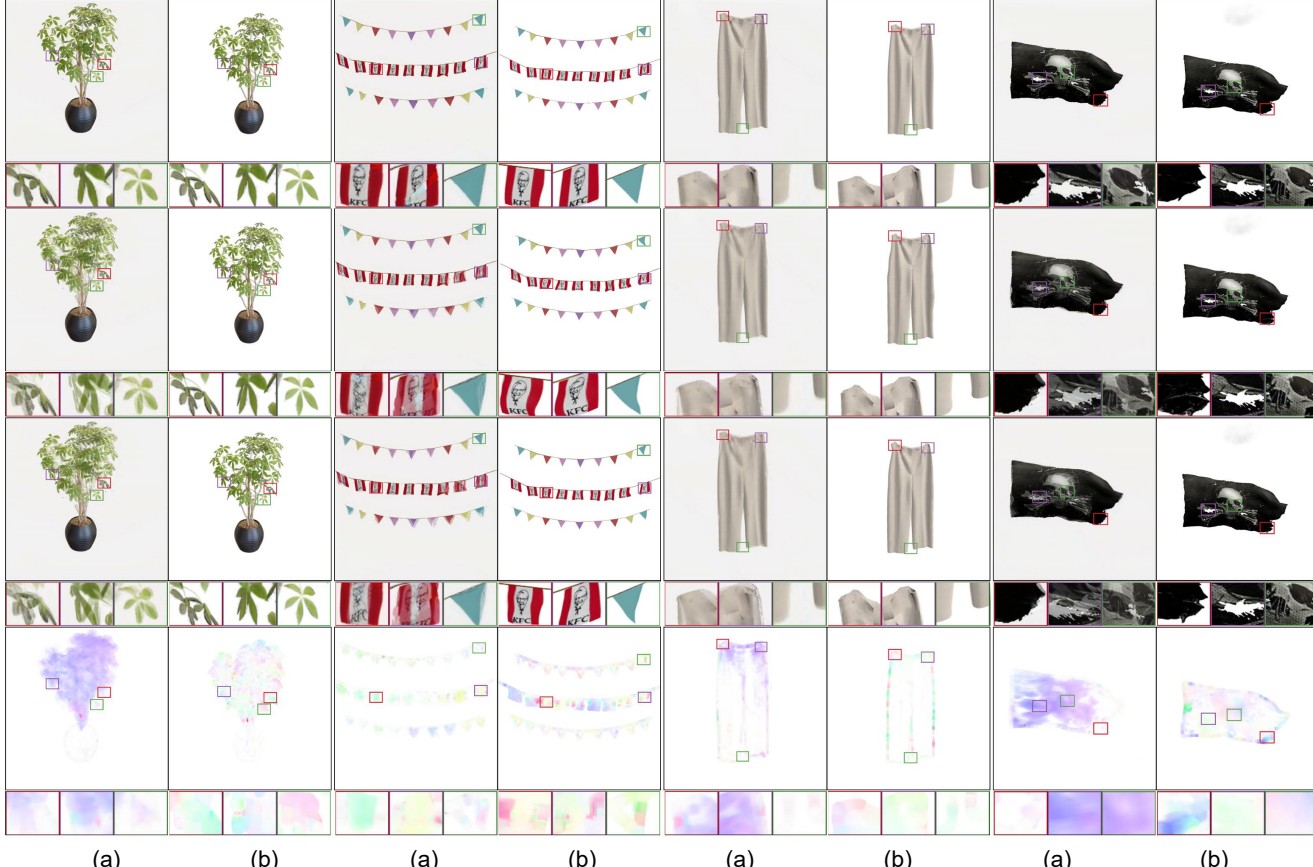

**Figure 3: Comparison of visual results.** From top to bottom, each column contains multiple key frames extracted from videos, and each screenshot accompanied by zoomed-in details. At the bottom, there is a visualization of the average optical flow map for the corresponding video, employing various colors to denote different motion directions. (a) is our method, and (b) is the method proposed by Li et al. [19].

optical flow maps closely resemble the reference optical flow maps, suggesting that our approach generates more realistic motion. We also use Fréchet Video Distance (FVD) [46] to evaluate the quality of the generated video. The comparison results can be seen in Table 2, which further demonstrates the advantage of our method.

***Qualitative Evaluation***. Visual comparisons between our proposed method and 3D Cinemagraph [19] are shown in Fig. 3. Despite being labeled as 3D Cinemagraph, their method primarily caters to fluid scenes and lacks precise geometric information in 3D space. Consequently, when handling geometrically continuous objects, their results can manifest severe artifacts, ultimately disrupting the scene's structure. In contrast, our method excels in preserving the object's geometric continuity, evident in the natural distortion of objects like a flag without tearing. Notably, as seen from the optical flow maps shown in the last row, their method typically features objects moving uniformly in one direction, whereas our approach demonstrates objects engaging in a periodic reciprocal motion, which is more aligned with real-world scenarios.

***User Study***. We conducted a user study involving 110 participants in answering which method produces videos that are more visually

**Table 3: User study on visual effects of generated videos.**

| Methods | User preference (%) |
|---------|---------------------|
| Li [19] | 5.77 |
| Ours | **94.23** |

realistic, and collected 104 valid questionnaires (excluding identical IPs). The comparison results are detailed in Table 3, showing that most participants preferred the results derived from our method.

## 5.4 Ablation Study

In this section, we conduct ablation studies to systematically analyze the impact of various components of our proposed method.

***Eccentricity Regularization***. As illustrated in Fig. 4, eccentricity regularization significantly reduces the occurrence of artifacts such as burrs and glitches in the scene during deformation. Sharp 3D Gaussians at the edges of the flag are prone to protrude during deformation, making it difficult for the originally connected points to remain in close proximity, consequently leading to the formation

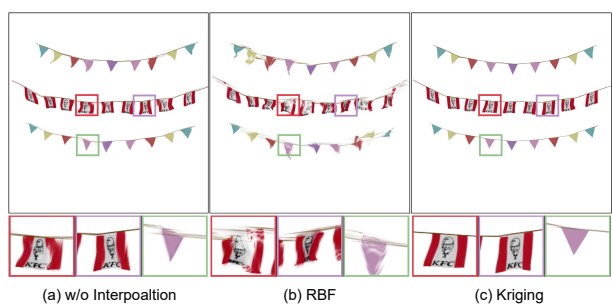

w/o eccentricity regularization    w/ eccentricity regularization

**Figure 4: Comparison of whether to use eccentricity regularization.** The use of the regularization term can significantly reduce the occurrence of burrs in the scene.

(a) w/o Interpoaltion    (b) RBF    (c) Kriging

**Figure 5: Comparison of different interpolation methods.** We compare the dense velocity fields obtained without interpolation (a), with RBF interpolation (b), and with Kriging interpolation methods, respectively.

of burrs. By incorporating eccentricity regularization, the shape of the 3D Gaussians is constrained to closely resemble a sphere, thereby alleviating this phenomenon.

***Interpolation Methods for Dense Velocity Vectors***. The choice of interpolation function for dense motion vectors directly affects the smoothness and accuracy of the motion field, which in turn influences the quality of the rendering result. The ablation results of interpolation is shown in Fig. 5. As can be seen, the objects are more complete and the motion of objects is more continuous when using Kriging interpolation, compared to no interpolation or RBF interpolation.

***Impact of Voxel Resolution on Clustering***. Voxel resolution directly impacts the granularity of spatial representation, thereby affecting the accuracy and granularity of cluster formation. As illustrated in Fig. 6, excessive resolution results in overly coarse scene division (Fig. 6a), leading to unrelated objects being grouped together, whereas overly small resolution causes excessive partitioning (Fig. 6c), potentially leading to loss of some higher-level information. We empirically adopt an appropriate voxel resolution that strikes a balance between scene segmentation and the preservation of scene information (Fig. 6b).

***Motion Amplitude Control***. The magnitude of motion for each 3D Gaussian point is controlled by $\psi$ in Eq. (12), with $\omega$ serving as

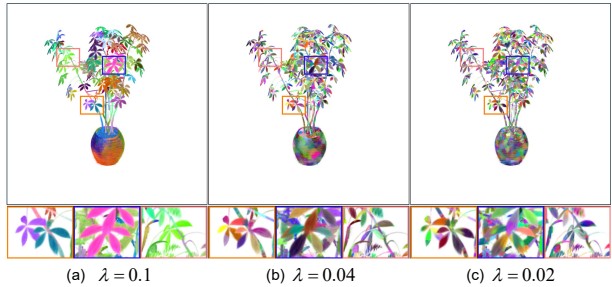

(a) $\lambda = 0.1$    (b) $\lambda = 0.04$    (c) $\lambda = 0.02$

**Figure 6: Clustering results at various voxel resolutions.** Distinct colors indicate different clusters. We aim to ensure that each individual object (e.g., a leaf) is encompassed within a single cluster (middle), rather than having multiple objects grouped into one cluster (left) or a single object fragmented across multiple clusters (right).

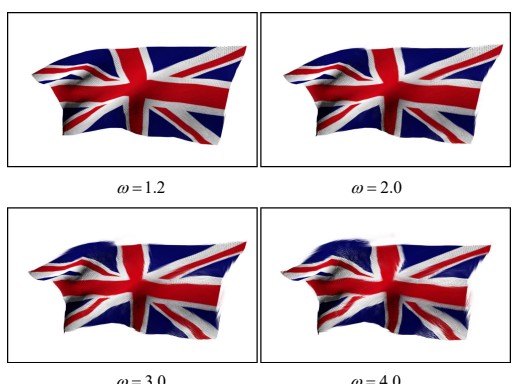

$\omega = 1.2$    $\omega = 2.0$

$\omega = 3.0$    $\omega = 4.0$

**Figure 7: Effect of the motion amplitude.** The deformation amplitude of the scene can be controlled by $\omega$. The larger $\omega$ is, the more intense the movement of the scene becomes. Note that excessively large values of $\omega$ may result in structural damage to the scene (lower right corner).

a hyperparameter to regulate $\psi$, and the impact of $\omega$ is shown in Fig. 7. A higher value of $\omega$ corresponds to a larger motion range for each 3D Gaussian point, resulting in more pronounced dynamics across the entire scene. However, it should be noted that excessively large values of $\omega$ may disrupt the continuity of the scene.

## 6 CONCLUSION

In this paper, we introduce LoopGaussian, a novel framework for generating authentic 3D cinemagraphs from multi-view images of static scenes. By leveraging 3D Gaussian Splatting and inherent scene self-similarity with an Eulerian velocity field, our method enables natural, loopable motion trajectories without extensive pre-training. LoopGaussian surpasses previous methods that are restricted to 2D image space, as we reconstruct the 3D geometry of the observed scene. Besides, our method enables rendering from any viewpoint and ensures consistency across multiple perspectives. Experiments demonstrate the effectiveness of our method in simulating motion for soft, non-rigid objects.

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

Received 20 February 2024; revised 12 March 2009; accepted 5 June 2009

