# OpenReview forum: "LoopGaussian: Creating 3D Cinemagraph with Multi-view Images via Eulerian Motion Field"
_acmmm.org/ACMMM/2024/Conference — MM2024 Oral_

### Official Review · Reviewer_HAmV · 2024-05-26

**Rating:** 3
**Confidence:** 3

**Summary:**

This paper reconstruction a static scene with 3DGS and use a regularization term to encourage ellipsoidal eccentricity.
Then the authors propose the concept of SuperGaussian to enhance the structure of Gaussians representaiton.
To animate the Gaussians, this paper distills the estimated sparse velocity field and dense velocity field into an MLP to help estimating the Eulerian motion field. Finally, the animation is obtained through interpolation.

**Strengths:**

This paper issues an interesting problem, which is like a 4D generation but is highly consistent with original scene.
Merely idea itself is novel enough for acceptance.
It's easy to agree that the geometric information leads to geometric consistency.
Methods are self-contained. And ablation study has a rich variety.

**Limitations:**

Lack of quantitative ablation study:
Although ablation study is rich, it lacks quantitative results for evaluaton.

Potential limited comparison in experiment:
The comparison is limited, with the compared method only takes one images as input, while this paper needs multi-view images for reconstruction.
Although this is not compulsory, it's better to compare with more recent methods with also multi-vew images as inputs, such as PhysGaussian.

Lack of video results in supplementary:
Since this paper proposes a methods with a loopable video as results. This is really uncompelling when no video results are submitted

**Suitability:**

2

---

### Official Review · Reviewer_Mf1G · 2024-05-26

**Rating:** 5
**Confidence:** 4

**Summary:**

The paper presents LoopGaussian, a novel method for creating 3D cinemagraphs by leveraging 3D Gaussian modeling and Eulerian motion fields. Cinemagraphs combine still photography with subtle motion to create visually captivating experiences. Traditional methods are limited to 2D image space and lack depth information. LoopGaussian addresses this limitation by reconstructing 3D Gaussian point clouds from multi-view images and employing shape regularization to prevent artifacts. The method then uses an autoencoder tailored for 3D Gaussian to project the point clouds into feature space, followed by clustering using SuperGaussian. By deriving an Eulerian motion field, the method animates the 3D Gaussian points, generating high-quality, seamlessly loopable 3D cinemagraphs. Experimental results demonstrate the effectiveness of this approach.

**Strengths:**

1. The proposed LoopGaussian method is highly innovative, effectively extending the concept of cinemagraphs from 2D to 3D space using 3D Gaussian modeling and Eulerian motion fields. This approach represents a significant advancement in the field.
2. The methodology is detailed and well-explained, with each step of the process clearly described. The use of shape regularization, the tailored autoencoder for 3D Gaussian, and the SuperGaussian clustering are well-justified and effectively implemented.
3. The experimental results are comprehensive and convincingly demonstrate the effectiveness of the proposed method. The generated 3D cinemagraphs are high-quality and visually appealing, showcasing the practical applicability of LoopGaussian.

**Limitations:**

1. This paper introduces a new dataset, however, the specific information about this data set is too little, such as what kind of scenes it mainly contains, how many scenes there are, etc.
2. What's the limitation of the proposed method?
3. What's the computational cost of the proposed method?

**Suitability:**

3

---

### Official Review · Reviewer_Hkm6 · 2024-05-30

**Rating:** 6
**Confidence:** 4

**Summary:**

This paper aims to generate cinemagraphs based on 3D scene, where the 3D scene is represented by multi-view images forming a 3D Gaussian point cloud. Cinemagraphs are a combination of motion and stillness, commonly used for dynamic presentations of 2D images. This paper's unique contribution is generating dynamic effects based on 3D static scene generation technology.

Additionally, I believe this paper not only implements the dynamic techniques commonly used in 2D images within a 3D scene but also addresses the issue of animating 3D Gaussian point clouds. This could have a significant impact and provide inspiration in the field.

**Strengths:**

1.Rich and precise vocabulary with clear and concise expression.
2.Provides new ideas for animating static 3D scenes.
3.Proposes a new method for scene presentation by creating cinemagraphs in 3D scenes.

**Limitations:**

1.The paper devotes too much description to the Gaussian spray technique. More space could be allocated to describing the contributions of this paper.
2.There are few comparisons with existing methods.
3.The paper does not discuss its own limitations.

**Suitability:**

3

---

### Meta-Review · Area_Chair_SXGk · 2024-07-04

**Recommendation:** Accept (Oral)
**Confidence:** 5

**Metareview:**

The paper presents a novel technique for creating animations from static scenes. The paper is well-written and the method is adequately described. Overall, it is a good contribution to ACM MM.
The presented LoopGaussian method is innovative (R1, R2, R3). The paper presents a new technique to animate static images using 3D Gaussian modeling and Eulerian motion fields. The LoopGaussian method is well explained and clearly presented. Using off-the-shelf components is well-advocated and justified (R1, R2, R3). The presented method is tested through adequate experiments. The results are convincing, and the animations are of good quality and visually pleasing. A comprehensive ablation study is also presented.
Most comments by the reviewers were addressed in the rebuttal, however, a couple of suggestions to further improve the paper were also made by the reviewers.  A comparison with more exiting similar techniques would justify the claims as suggested by reviewer R1. Moreover, if there are any limitations of the presented method, they must be discussed (R1, R2).